# Preliminary Study on the Inhibitory Effect and Mechanism of *Eichhornia crassipes* on Co-Cultured *Raphidiopsis raciborskii*

Xiaowei Cheng [1], Weibin Pan [1,2,3,*], Yuanyuan Hu [1], Yulin Zou [1] and Xiaojia Huang [1]

1 School of Environment and Energy, South China University of Technology, Guangzhou 510006, China; 201820134722@mail.scut.edu.cn (X.C.)
2 Key Laboratory of Pollution Control and Ecosystem Restoration in Industry Clusters, Ministry of Education, Guangzhou 510006, China
3 Engineering and Technology Research Centre of Guangzhou Water Resource and Water Environment, Guangzhou 510006, China
* Correspondence: ppwbpan@scut.edu.cn; Tel.: +86-139-2220-5690

**Abstract:** *Raphidiopsis raciborskii*, which harms water supply and ecological security, may expand its range with climate warming. To explore the ecological management method of *R. raciborskii* blooms, the co-culture system of *Eichhornia crassipes* and *R. raciborskii* was established to study the inhibitory effect and inhibition mechanism of *E. crassipes* on *R. raciborskii* compared with the shaded culture of *R. raciborskii* alone. It was found that after co-cultured with *E. crassipes* (fresh weight, $5.10 \pm 0.60$ g·L$^{-1}$) for seven days under initial high nutrient concentration, the algal cell density of co-culture groups with an initial cell density of $1.57 \times 10^8$ cells·L$^{-1}$ and $1.57 \times 10^9$ cells·L$^{-1}$ decreased by 63.19% and 14.87%, respectively. Meanwhile, the algal cell density in the co-culture group was significantly lower than that in the control group ($p < 0.01$). The algal cell density change showed that the growth of *R. raciborskii* was inhibited by co-cultured *E. crassipes*. In addition, *E. crassipes* could significantly reduce the nitrogen and phosphorus content in the water, but the growth state and photosynthesis of *E. crassipes* were not inhibited by *R. raciborskii*. These results suggested that *E. crassipes* has the potential to control *R. raciborskii* blooms. The study also found that when co-cultured, there was no significant difference in algal cell density between the groups supplemented with a certain amount of nutrients and the group without additional nutrients. Meanwhile, the inhibition rates of the co-culture groups were over 99% on day 4. These results showed that nutrient factors had no significant effect on the inhibitory effect of *E. crassipes* in the range of nutrients regulated by the experiment, and other factors played a leading role. In other words, under experimental conditions, nutrient competition was not the primary inhibitory mechanism of *E. crassipes*. This study also confirmed that *E. crassipes* had inhibitory allelopathy on *R. raciborskii*. The contribution and significance of allelopathy and other mechanisms could be studied in the future.

**Keywords:** allelopathy; cyanobacteria blooms; *Eichhornia crassipes*; interspecific relationship; nutrient competition; *Raphidiopsis raciborskii*

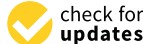



## 1. Introduction

Eutrophication caused by excessive nitrogen and phosphorus nutrient input has become a global environmental concern. Algal bloom, especially cyanobacteria bloom, is one of the most severe symptoms of lake eutrophication, endangering the safety of water resources and harming ecosystems [1,2]. *Raphidiopsis raciborskii* (formerly *Cylindrospermopsis raciborskii*), a freshwater filamentous cyanobacterium, was first identified as a harmful bloom species after a significant human poisoning incident on Palm Island in 1979 [3]. *R. raciborskii* was originally distributed in tropical and subtropical regions, but in recent decades, researchers have found that it has spread to temperate regions [4,5]. Global warming and the excellent environmental adaptability of *R. raciborskii* may be the critical reasons for the spreading [4,6]. *R. raciborskii* has flexible nutrient uptake strategies, which

makes it tolerant to a low-nutrient environment. It fixes nitrogen by forming heteromorphic cells and utilizes various types of phosphorus and nitrogen sources [7]. In addition, it has a wide range of adaptations to light, temperature, and other environmental factors; therefore, it can withstand adverse conditions, such as shade and low temperature [8–12]. *R. raciborskii* also inhibits co-cultured algae growth by releasing allelochemicals [13]. In some tropical and subtropical areas, *R. raciborskii* has replaced *Microcystis* to become the main cyanobacteria endangering the safety of the reservoir water supply in the region [14,15]. Under the continuous drive of global warming and water eutrophication, its harmful scope will continue to expand, and it is urgent to seek corresponding control methods. However, existing studies on the control of algal blooms mainly focus on *Microcystis* and *Anabaena*, which were common past algal blooms. In contrast, studies on *R. raciborskii* are still scarce.

Chemical algaecides, such as copper sulfate, are effective and easy to operate. However, they usually have adverse effects on aquatic organisms and cannot fundamentally solve the problem of eutrophication [16]. To find an ecofriendly and sustainable method for controlling *R. raciborskii* blooms, we turned our attention to biological methods. Early in 1949, Hasler et al. demonstrated through field experiments that aquatic macrophytes could inhibit phytoplankton growth [17]. Aquatic macrophytes can inhibit algae growth by shading, nutrient competition, releasing inhibitory allelochemicals, remodeling habitats, and so on [18–21]. However, little is known about using aquatic macrophytes to control *R. raciborskii*.

*Eichhornia crassipes*, an aquatic macrophyte native to tropical South America, was originally spread around the world as an ornamental plant. *E. crassipes* grows and reproduces rapidly and has a wide range of environment adaptability, which will damage the aquatic ecosystem when it is propagated in the wild [22,23]. *E. crassipes* has been regarded as an invasive plant in many countries, and its control measures have been widely studied [24]. Eliminating *E. crassipes* completely is difficult, but researchers have found that *E. crassipes* has tremendous absorption capacity and strong environmental tolerance, which can be used for phytoremediation and water purification [25–27]. *E. crassipes* is widely used in the ecological restoration of eutrophic water and has proven to be effective [28,29]. Many aquatic macrophytes cannot survive and reproduce in severely eutrophic or algal bloom water, but *E. crassipes* can grow well in water filled with thick cyanobacterial scum [30,31]. Recent studies have indicated that *E. crassipes* can inhibit the growth of *Scendesmus obliquus*, *Chlorella vulgaris*, *Microcystis aeruginosa*, and so on [32–34]. For *R. raciborskii*, researchers have found that allelochemicals, such as N-phenyl-2-naphthylamine and linoleic acid, in the roots of *E. crassipes* can inhibit the growth of *R. raciborskii* [35–37]. *E. crassipes* may be a helpful tool for phytoremediation of eutrophic water bodies with *R. raciborskii* bloom. However, to the best of our knowledge, there has yet to be a study on the interspecific relationship between *E. crassipes* and *R. raciborskii*.

To determine whether *E. crassipes* has the potential to control *R. raciborskii* blooms, in this study, we constructed a co-culture system of *E. crassipes* and *R. raciborskii* and regulated the nutrient concentration and the algal cell density in the system. This study aimed to observe the interaction between *E. crassipes* and *R. raciborskii* to verify whether *E. crassipes* can effectively inhibit the growth of *R. raciborskii* and further explore the inhibition mechanism of *E. crassipes*. Because *E. crassipes* has an outstanding ability to absorb nutrients, which are essential factors in regulating algae growth, we initially hypothesized that nutrient competition is an important mechanism of inhibition. This study can provide a basis for effectively using *E. crassipes* and other aquatic macroplants to control *R. raciborskii* blooms.

## 2. Materials and Methods

### 2.1. Experimental Material Preparation

#### 2.1.1. E. crassipes

The *E. crassipes* used in the experiment came from a pond (23°36′11.35″ N, 113°33′25.02″ E) in Guangzhou, China. The roots and leaves of the *E. crassipes* were brushed and rinsed repeatedly with tap water to remove the attached algae and sediment. Then, the washed

*E. crassipes* plants were placed in plastic basins containing nutrient solutions and cultured under natural light. The nutrient solution was replaced, and the withered roots and leaves were cut off every 10 days. Before the co-culture experiment, healthy and similarly growing *E. crassipes* plants were repeatedly washed with pure water and then pre-cultured in a self-made illumination cultivation frame for two days.

### 2.1.2. *R. raciborskii*

Cultures of *R. raciborskii* (FACHB 1503) were procured from the Freshwater Algae Culture Collection at the Institute of Hydrobiology, Chinese Academy of Sciences (Wuhan, China). The cultures were aseptically cultivated in sterilized BG11 medium and cultivated at 25 °C under a 12 h:12 h light–dark cycle with an illumination intensity of 2500~3000 lux in an illuminating incubator (LRH-400-G, Guangdong Taihongjun Scientific Instrument Co., Shaoguan, China). The cultures were manually shaken twice daily to prevent adherent growth and precipitation of algal cells. The algae were expanded every 14~28 days, and the cells in the exponential growth period were prepared for the experiments.

### 2.1.3. Self-Made Illumination Cultivation Frame

We set up an illumination cultivation frame in the laboratory for the co-culture experiments (Figure 1). The indoor temperature was controlled to be higher than 21 °C by air conditioning, and the humidity range was 37~76%. A six-layer stainless steel frame (50 cm × 40 cm × 175 cm) was prepared for self-design. The top of each layer of the frame was evenly provided with five cold white light-led tubes in series, and a timing switch was connected to control the light–dark cycle to be 14 h:10 h. The light intensity of this frame ranged from 3000~8000 lux. During the experiment, the position of the culture container of each experimental group was changed regularly to avoid cultural differences caused by different illumination.

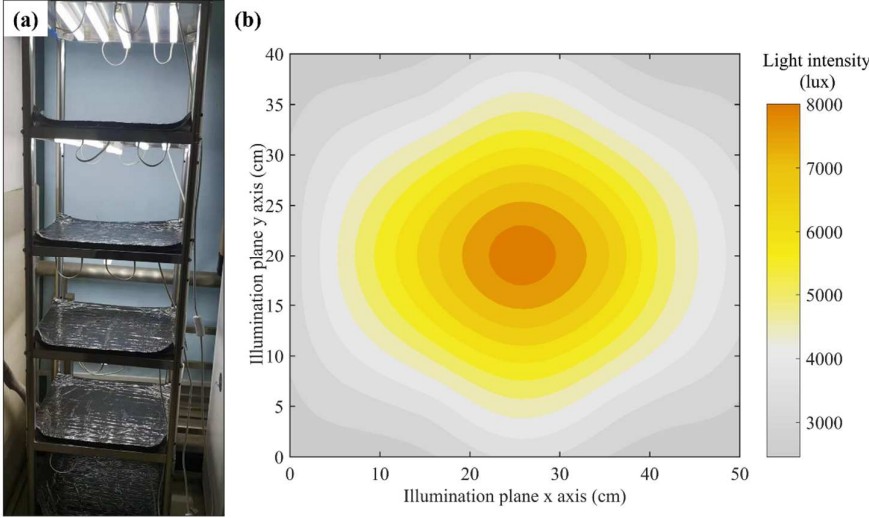

**Figure 1.** Self-made illumination cultivation frame (**a**) and its light intensity (**b**).

### 2.2. *Experimental Design*

### 2.2.1. Experiment 1: Co-Culture of *E. crassipes* and *R. raciborskii* with Different Densities

To investigate whether *E. crassipes* had an inhibitory effect on *R. raciborskii*, according to the previous report [38–40], three different initial algal cell densities of *R. raciborskii* and *E. crassipes*, with an initial fresh weight of 5.10 ± 0.60 g, were co-cultured in a beaker containing 1 L culture solution. At the same time, the single culture of *R. raciborskii* (R-Group) and the single culture of *E. crassipes* (E-Group) were set as controls.

During *R. raciborskii* bloom, the algal cell density usually reaches above $1.50 \times 10^7$ cells·L$^{-1}$ [41,42]. To simulate the bloom state of *R. raciborskii*, the initial algal cell

densities of the three co-culture treatment groups were $1.57 \times 10^9$ cells·$L^{-1}$ (HRE-Group), $1.57 \times 10^8$ cells·$L^{-1}$ (MRE-Group), and $1.57 \times 10^7$ cells·$L^{-1}$ (LRE-Group), respectively. In the R-Group, the shading effect of *E. crassipes* was simulated by pasting 6 cm-wide black card paper, and the initial algal cell density was set to $1.57 \times 10^9$ cells·$L^{-1}$. All groups set four replicates.

The culture medium of this experiment was based on 1/10 BG11 medium, and 27 mg $K_2SO_4$ and 14 mg $CaCl_2 \cdot 2H_2O$ were added to supplement the potassium and calcium elements needed for the growth of higher plants.

Incubation lasted for seven days. Aliquots (4 mL) from each culture were sampled daily to observe the density and morphology of the algal cells. The chlorophyll fluorescence of *E. crassipes*, the pH, and the conductivity of the cultures were observed in situ every other day. Before and after the culture, the fresh weight, plant height, root length, and leaf number of *E. crassipes* were measured. Thirty milliliter aliquots of the cultures were sampled to measure the concentration of total dissolved phosphorus (TDP) and total dissolved nitrogen (TDN).

### 2.2.2. Experiment 2: Nutrient Enrichment Experiment

According to previous studies with some modification, we designed the nutrient enrichment experiment to assess whether the nutrient factors would affect the relationship between *E. crassipes* and *R. raciborskii* and further analyze the inhibition mechanism of *E. crassipes* [43,44]. Two monoculture control groups of *R. raciborskii* and three co-culture treatment groups were set up, and then their nutrient conditions were regulated. In the monoculture groups, one group was not supplemented with nutrients (mo-Group), and the other group was supplemented with sufficient phosphorus (mo-P-Group). In the co-culture groups, one group was not supplemented with nutrients (co-Group); another group was supplemented with sufficient phosphorus (co-P-Group); and the other group was supplemented with sufficient nitrogen, potassium, calcium, and other nutrients except phosphorus contained in BG11 medium (co-N, K, Ca, et-Group).

Each group was grown in a polypropylene plastic box containing 1.5 L of modified BG11 medium formulated as follows: all the components except phosphate and nitrate were in the same concentration as BG11 medium, and concentrations of phosphate and nitrate were decreased to 0.24 mg·$L^{-1}$ and 5.56 mg·$L^{-1}$, respectively. The initial algal cell density of *R. raciborskii* was $3.39 \times 10^8$ cells·$L^{-1}$, and the initial fresh weight of co-cultured *E. crassipes* was $8.05 \pm 1.93$ g. The shading effect of *E. crassipes* was simulated by pasting 7 cm-wide black card paper in the monoculture group. All groups set four replicates.

To avoid the addition of other non-regulatory elements when the corresponding nutrients were supplemented, the supplementary phosphorus source was $Na_2HPO_3$ solution, the nitrogen source was $KNO_3$ solution, and the potassium source was $K_2SO_4$ solution. Calcium, magnesium, iron, and trace elements were supplemented by preparing the corresponding mother liquor and A5 trace element solutions. The supplementation of nutrients in each group during culture is shown in Table 1.

**Table 1.** The supplementation of nutrients in each group (the initial volume of the culture solution was 1.5 L). "–" in the table indicate no addition.

| Time | Nutrient Supplement Amount in Each Group (mg) | | | | |
|------|--------------|--------------|----------|------------|---------------------|
| | **mo-Group** | **mo-P-Group** | **co-Group** | **co-P-Group** | **co-N, K, Ca, et-Group** |
| 0 h | P: 0.36 mg, N: 8.3 mg, K: 20 mg, Ca: 15 mg, Mg: 11 mg, Fe: 1.9 mg, B: 0.75 mg, Mn: 0.77 mg, Mo: 0.23 mg, Zn: 0.075 mg, Cu: 0.031 mg, Co: 0.015 mg | | | | |
| 12 h | – | P: 0.09 | – | P: 0.26 | N: 0.70 |

**Table 1.** *Cont.*

| Time | Nutrient Supplement Amount in Each Group (mg) | | | | |
|---|---|---|---|---|---|
| | mo-Group | mo-P-Group | co-Group | co-P-Group | co-N, K, Ca, et-Group |
| 24 h | – | P: 0.50 | – | P: 0.64 | N: 4.0, K: 2.0, Ca: 1.5, Mg: 0.74, Fe: 0.19, B: 0.075, Mn: 0.077, Mo: 0.023, Zn: 0.0075, Cu: 0.0031, Co: 0.0015 |
| 48 h | – | P: 0.31 | – | P: 0.93 | – |
| 72 h | – | P: 0.19 | – | P: 0.57 | N: 0.70, K: 2.0, Ca: 1.5, Mg: 0.74, Fe: 0.19, B: 0.075, Mn: 0.077, Mo: 0.023, Zn: 0.0075, Cu: 0.0031, Co: 0.0015 |

Incubations lasted for seven days. Samples were taken at specific time intervals to observe the growth of *R. raciborskii* and to determine the concentrations of TDP and TDN in the culture solution. Purified water was added at 40 mL daily to supplement the reduced water from evaporation and sampling.

2.2.3. Experiment 3: Exposure to *E. crassipes* Exudates

This experiment was conducted to investigate whether *E. crassipes* exudates have an inhibitory effect on *R. raciborskii*. A healthy *E. crassipes* (fresh weight, 4.60 g) plant was cultured in a beaker filled with 1 L of pure water and cultured on the self-made illumination cultivation frame for four days. Exudates were obtained by filtering *E. crassipes* planting water through the 0.22 μm mixed fiber microfiltration membrane [45]. Then, BG11 medium was prepared with *E. crassipes* exudates, and a control group was prepared with pure water. The prepared BG11 medium was put into a 500 mL conical flask, and 300 mL of BG11 medium and 30 mL of *R. raciborskii* cultures were added to each flask. The conical flasks were then placed into an illuminating incubator. The culture conditions were the same as in Section 2.1.2. All groups set three replicates. The algal cell density of *R. raciborskii* was measured by taking 4 mL of the samples every other day.

*2.3. Observations of the Growth of R. raciborskii*

After being stained with Lugol's solution, *R. raciborskii* cells were observed under a microscope (BX41; Olympus Corporation, Tokyo, Japan) equipped with an algal auxiliary identification system (Algacount R100, Shineso Co., LTD, Hangzhou, China) and a phytoplankton counting box (CC-F 0.1 mL, Purity Instrument Co., LTD, Beijing, China). Since it is difficult to distinguish individual algal cells of *R. raciborskii* at 200 times and 400 times magnification, we first measured the average length of individual algal cells at 1000 times magnification and then measured the length of trichomes at 200 times or 400 times magnification to estimate algal cell density. The length of each trichome was measured using ImageJ software (download from https://imagej.nih.gov/ij/index.html (accessed on 13 February 2021)) after each field of view was photographed and stored using Algacount R100.

The relative inhibition rate (IR) was calculated from the algal cell density following each of the experimental treatments and the control treatment using the following equation:

$$IR\ (\%) = (1 - N_{t\text{-treatment}} / N_{t\text{-control}}) \times 100 \tag{1}$$

where $N_{t\text{-control}}$ and $N_{t\text{-treatment}}$ represent the algal cell density of the experimental treatments and the control treatment groups, t (day) denotes the co-culture time.

### 2.4. Measurement of Growth and Chlorophyll Fluorescence of E. crassipes

The growth characteristics included fresh weight, plant height, root length, and leaf number. When determining the fresh weight, the water in the roots and leaves of *E. crassipes* was absorbed by filter paper and then weighed using an analytical balance. The plant height and root length were measured with a ruler. The leaf number was calculated with a standard leaf length exceeding 2 cm.

To characterize the photosynthetic features of *E. crassipes*, chlorophyll fluorescence at the midpoint of the leaves (six points for each replicate) was measured using a pulse amplitude-modulated fluorometer (Junior PAM, Heinz Walz GmbH, Effeltrich, Germany) after 20 min of dark acclimation. The maximum photochemical quantum yield of PSII (Fv/Fm) was calculated as follows:

$$Fv/Fm = (Fm - Fo)/Fm \qquad (2)$$

where Fm and Fo represent the maximal fluorescence and the baseline fluorescence, respectively. *E. crassipes* leaves were irradiated with nine actinic lights (0, 66, 99, 125, 190, 285, 420, 625, and 845 µmol photon $m^{-2}$ $s^{-1}$) of gradually increasing intensity by Junior PAM, and the fluorescence response after irradiation was detected to obtain rapid light curves (RLCs) [46,47]. The light utilization efficiency (α) and maximal relative electron transport rate (rETRmax) were derived from RLCs and were calculated automatically by WinControl V3.25 supporting software.

### 2.5. Determination of Water Quality Parameters

The pH and conductivity were assessed using a portable pH meter (ST300, Ohaus Instruments Co., LTD, Changzhou, China) and a conductivity meter (DDS-11A, Shanghai INESA Scientific Instrument Co., LTD, Shanghai, China), respectively. TDP was determined using the ammonium molybdate spectrophotometric method, and TDN was determined using the alkaline potassium persulfate digestion UV spectrophotometric method [48].

### 2.6. Statistical Analysis

Excel 2016 software (Microsoft, Redmond, WA, USA) was used for data aggregation, and Origin Pro 2018 C software (Origin Lab, Northampton, PA, USA) was used for drawing. Statistical analyses were performed using SPSS 22.0 software (IBM, Armonk, NY, USA). Statistical differences in *R. raciborskii* algal cell densities of the control group and experimental group in Experiment 1 and Experiment 3 were assessed using two independent sample t-tests, as was the concentration of TDP and TDN in the culture medium before and after co-culture in Experiment 1. One-way analysis of variance (ANOVA), followed by the least significant difference test, was used to compare the differences in the growth and photosynthetic characteristics of *E. crassipes*, the changes in pH and conductivity in the culture medium in Experiment 1, and the changes in *R. raciborskii* algal cell density under different treatments in Experiment 2. The significance level was set to 0.05.

## 3. Results

### 3.1. Interaction between E. crassipes and R. raciborskii in Co-Culture

In Experiment 1, the cell density, trichome size of *R. raciborskii*, and changes in the growth and photosynthetic characteristics of *E. crassipes* in the co-culture system were mainly observed to analyze whether *E. crassipes* has the potential to control the growth of *R. raciborskii*.

#### 3.1.1. Effects of E. crassipes on the Growth and Trichome Size of R. raciborskii

When *R. raciborskii* was cultured alone (R-Group), the algal cell density increased during the experiment (Figure 2a). However, the trend of algal cell density was different when co-cultured with *E. crassipes*. The algal cell densities of the HRE-Group and the MRE-Group began to decrease on day 5 (Figure 2a), and they decreased by 14.87% and 63.19%,

respectively, compared with the initial density on day 7. On the contrary, the algal cell density of the LRE-Group began to decrease on day 7, so it did not fall in numbers during the experiment (Figure 2a). Although the algal cell density of the HRE-Group increased continuously from day 0 to day 4, it was significantly lower than that of the R-Group from day 3 ($p < 0.01$). On day 7, the IR of the HRE-Group reached 66.28% (Figure 2a,b).

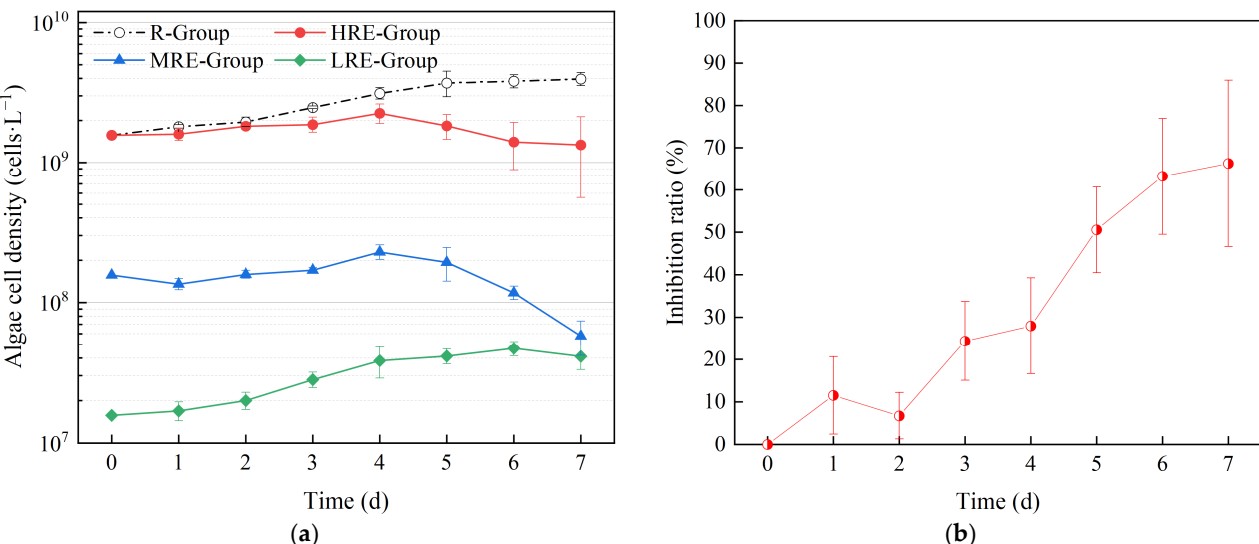

(**a**)           (**b**)

**Figure 2.** (**a**) The algal cell density of *R. raciborskii* co-cultured with *E. crassipes* at different initial densities (mean ± SD, *n* = 4); (**b**) inhibition rate of *R. raciborskii* with high initial algal cell density during co-culture with *E. crassipes* (HRE-Group) (mean ± SD, *n* = 4).

In general, the trichome length of *R. raciborskii* was widely distributed (Figure 3). The median trichome length in each group was close to the mean but slightly less than the mean, indicating that the trichome length was concentrated on the relatively shorter side (Figure 3). During the experiment, the development trend of trichome length was different between cultured alone and co-cultured with *E. crassipes*. In the R-Group, the average length and distribution range of the algal trichomes gradually increased (Figure 3a). In comparison, the average length of algal trichomes in the three co-culture groups showed no apparent trend of lengthening (Figure 3b–d). After seven days of co-culture, the average trichome lengths of HRE-Group, MRE-Group, and LRE-Group were 102.9 µm, 94.8 µm, and 137.7 µm, respectively, which were shorter than that of the R-Group, 145.5 µm, indicating that the presence of *E. crassipes* may inhibit the lengthening of *R. raciborskii* trichomes (Figure 3).

### 3.1.2. Effects of *R. raciborskii* on the Growth and Photosynthesis of *E. crassipes*

After a seven-day incubation, the fresh weight, plant height, root length, and leaf number of *E. crassipes* in the co-culture group increased. The increment of these indicators had no significant difference with *E. crassipes* cultured alone (Table 2).

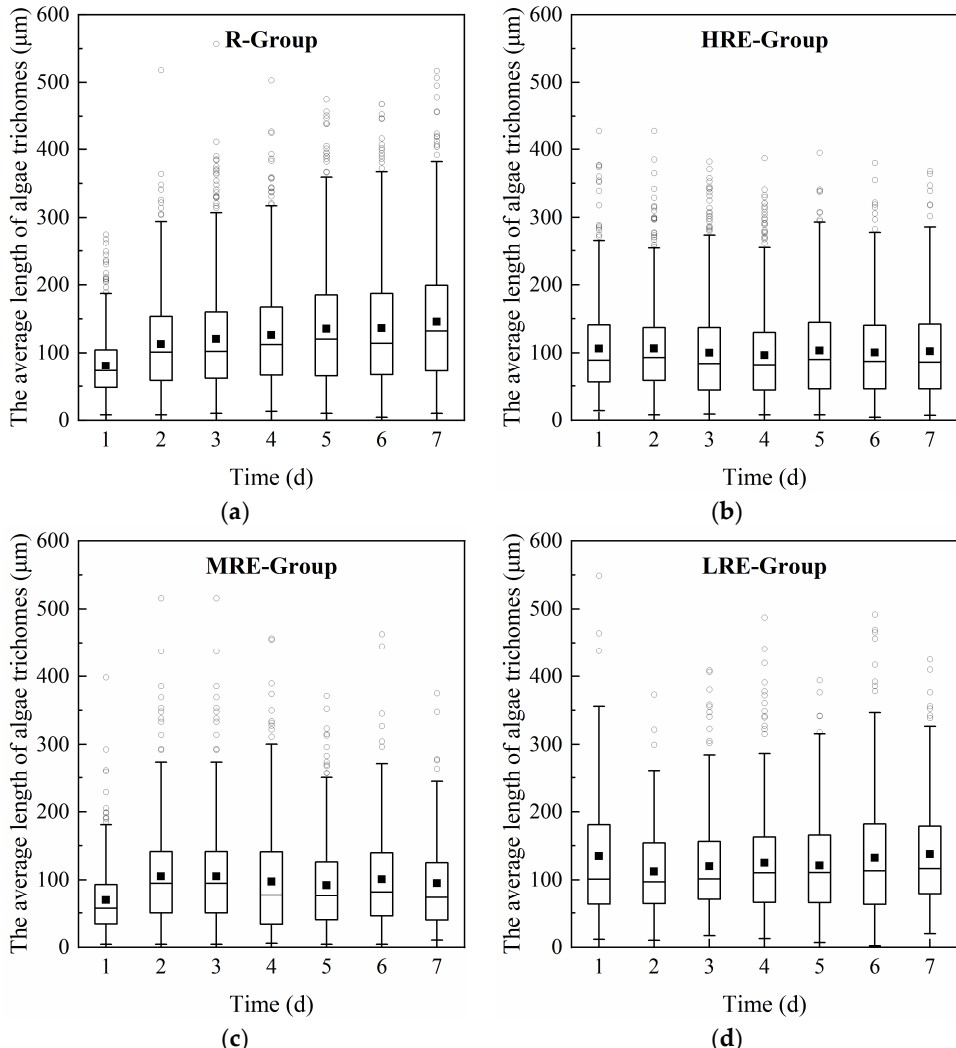

**Figure 3.** Frequency distribution of *R. raciborskii* trichome length in (**a**) the R-Group, (**b**) the HRE-Group, (**c**) the MRE-Group, and (**d**) the LRE-Group. In the box plots, the solid line in the middle represents the median, the filled square represents the mean, and the lower and upper box lines represent the lower and upper quartiles.

**Table 2.** Changes in the growth characteristics of *E. crassipes* before and after co-culture. Values are mean $\pm$ SD (*n* = 4). Different lowercase letters indicate significant differences between groups at $p < 0.05$.

| Growth Indicator | Groups | | | |
|---|---|---|---|---|
| | E-Group | HRE-Group | MRE-Group | LRE-Group |
| fresh weigh increment (g) | 3.54 $\pm$ 0.74 [ab] | 4.36 $\pm$ 1.11 [a] | 3.53 $\pm$ 1.15 [ab] | 2.65 $\pm$ 0.31 [b] |
| plant height increment (cm) | 1.05 $\pm$ 0.79 [a] | 0.55 $\pm$ 0.06 [a] | 0.63 $\pm$ 0.42 [a] | 0.85 $\pm$ 0.50 [a] |
| root length increment (cm) | 0.63 $\pm$ 0.30 [a] | 1.00 $\pm$ 1.04 [a] | 0.68 $\pm$ 1.04 [a] | 0.85 $\pm$ 0.83 [a] |
| leaf number increment | 3.00 $\pm$ 1.15 [a] | 2.75 $\pm$ 0.96 [a] | 4.50 $\pm$ 2.08 [a] | 3.00 $\pm$ 2.00 [a] |

On day 6, the light energy utilization efficiency ($\alpha$) and the maximum photochemical quantum yield (Fv/Fm) of the HRE-Group were significantly higher than the initial values and the values of E-Group and LRE-Group ($p < 0.01$; Table 3). In addition, compared with the group *E. crassipes* cultured alone (E-Group), the $\alpha$ value and the Fv/Fm value of the MRE-Group and the LRE-Group showed no significant difference on days 3 and 6 (Table 3). These results indicate that the photosynthetic capacity of *E. crassipes* was not inhibited by

*R. raciborskii* in this experiment. After six days of co-culture, the Fv/Fm values of the three co-culture groups were significantly higher than the initial values ($p < 0.01$), while there was no significant difference in the *E. crassipes* cultured alone, indicating that co-culture with *R. raciborskii* may promote the photosynthesis of *E. crassipes* to a certain extent (Table 3).

**Table 3.** The efficiency of electron transport ($\alpha$) and maximum photochemical yield (Fv/Fm) of *E. crassipes* leaves during co-culture. Values are mean $\pm$ SD ($n = 4$). Different capital letters indicate significant differences at different time points in the same group; different lowercase letters indicate significant differences between groups at the same time point. The significance difference level was set to 0.05.

| Parameters | Time | Groups | | | |
|---|---|---|---|---|---|
| | | E-Group | HRE-Group | MRE-Group | LRE-Group |
| $\alpha$ | Day 0 | 0.192 $\pm$ 0.014 [Aa] | 0.212 $\pm$ 0.028 [Aa] | 0.191 $\pm$ 0.038 [Aa] | 0.212 $\pm$ 0.014 [Aa] |
| | Day 3 | 0.235 $\pm$ 0.012 [Aa] | 0.231 $\pm$ 0.018 [ABa] | 0.232 $\pm$ 0.009 [Aa] | 0.224 $\pm$ 0.017 [Aa] |
| | Day 6 | 0.209 $\pm$ 0.014 [Aa] | 0.261 $\pm$ 0.020 [Bb] | 0.194 $\pm$ 0.026 [Aa] | 0.191 $\pm$ 0.017 [Aa] |
| Fv/Fm | Day 0 | 0.815 $\pm$ 0.041 [Xx] | 0.814 $\pm$ 0.020 [Xx] | 0.818 $\pm$ 0.024 [Xx] | 0.809 $\pm$ 0.021 [Xx] |
| | Day 3 | 0.846 $\pm$ 0.011 [Xxy] | 0.827 $\pm$ 0.004 [Xz] | 0.852 $\pm$ 0.009 [XYx] | 0.836 $\pm$ 0.005 [Yyz] |
| | Day 6 | 0.849 $\pm$ 0.011 [Xx] | 0.902 $\pm$ 0.024 [Yy] | 0.881 $\pm$ 0.038 [Yxy] | 0.849 $\pm$ 0.004 [Yx] |

### 3.1.3. Changes in Nutrients and Other Water Quality Factors

After the culture, compared with the control group R-Group, the concentrations of TDP and TDN in the three co-culture groups decreased significantly ($p < 0.01$), and the concentration of TDP was close to 0 mg/L (Figure 4a,b). The presence of *E. crassipes* accelerated phosphorus consumption in the water.

The changes in pH and conductivity of each group are shown in Figure 4c,d. Overall, the pH and conductivity changes of the three co-culture groups were similar. The initial pH and conductivity of each group were slightly different, mainly due to the different amounts of algae liquid added at the beginning of the experiment. The pH value of the R-group increased gradually with the culture time, and it was as high as 9.78 on day 6, which was significantly higher than that of the other three co-culture groups ($p < 0.01$; Figure 4c). The results showed that the presence of *E. crassipes* reduced the pH value of the water relatively and made it neutral. In addition, from day 2, the conductivity of the three co-culture groups was significantly lower than that of the control group ($p < 0.01$; Figure 4d), which means that the presence of *E. crassipes* reduced conductivity relatively. The conductivity fluctuation during the experiment might be related to changes in environmental temperature.

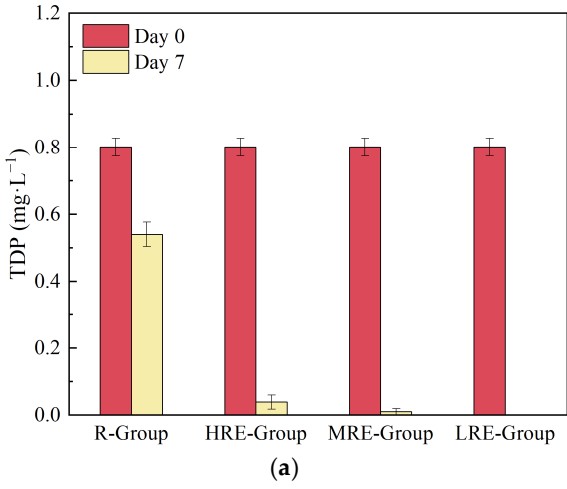

(a)

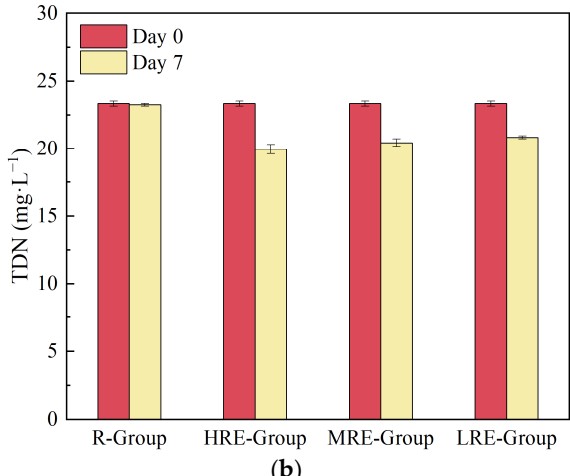

(b)

**Figure 4.** *Cont.*

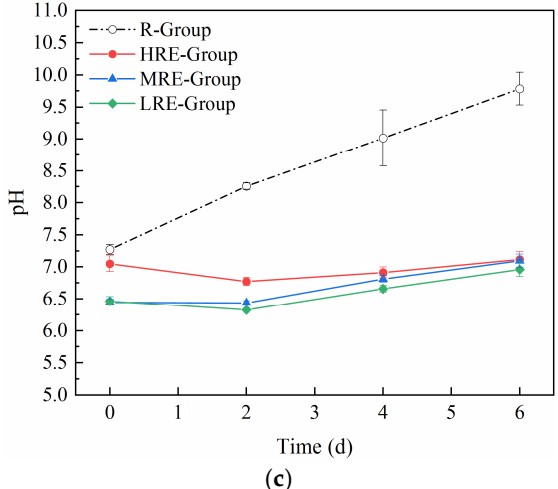
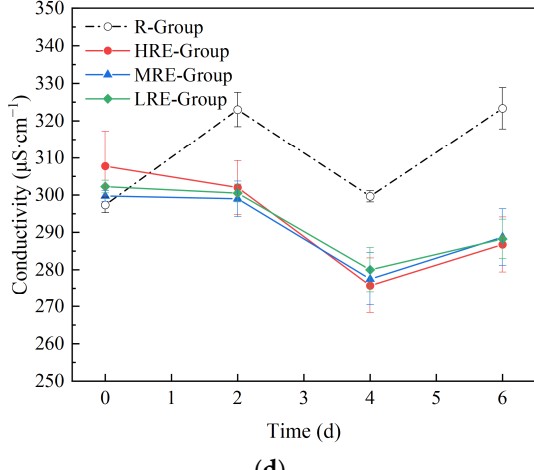

(**c**) (**d**)

**Figure 4.** The concentrations of (**a**) TDP and (**b**) TDN, and the variation of (**c**) pH and (**d**) conductivity in the culture medium during co-culture with *E. crassipes* (mean ± SD, *n* = 4).

### 3.2. Effects of Nutrients on Algae Inhibition of E. crassipes

In Experiment 2, the addition of nutrients was adjusted, and the inhibitory effect of *E. crassipes* on *R. raciborskii* was observed to explore whether nutrient competition is an essential mechanism of *E. crassipes* inhibiting the growth of *R. raciborskii*.

When *R. raciborskii* was cultured alone, the phosphorus in the culture medium was almost exhausted by day 2 if it was not supplemented (mo-Group; Figure 5a). However, the algal cell density of the mo-Group continued to increase until day 6 (Figure 5e). Nevertheless, compared with the phosphorus supplementation treatment (mo-P-Group), the algal cell density of the mo-Group was significantly lower than that of the mo-P-Group from day 3 ($p < 0.01$; Figure 5e), and the IR was 58.04% on day 7 (Figure 5f). These results indicate that phosphorus is a limiting factor for the growth of *R. raciborskii* under experimental conditions, and the relative deficiency of nutrients inhibits the growth of *R. raciborskii*.

When *R. raciborskii* was co-cultured with *E. crassipes*, one group was not supplemented with nutrients (co-Group); another group was supplemented with sufficient phosphorus (co-P-Group); and the other group was supplemented with sufficient nitrogen, potassium, calcium, and other nutrients except phosphorus contained in the BG11 medium (co-N, K, Ca, et-Group). The supplement of phosphorus and nitrogen is shown in Figure 5a–d. During the experiment, the concentration of TDP ranged from 0.07 to 1.09 mg·L⁻¹ in the co-P-Group, and the concentration of TDN ranged from 1.61 to 7.27 mg·L⁻¹ in the co-N, K, Ca, et-Group (Figure 5a,b).

Although nutrient supplementation was disparate in the three co-culture groups, their algal cell densities had a consistent trend. After two days of co-culture, the algal cell densities of co-Group; cos-P-Group; and co-N, K, Ca, et-Group decreased sharply, and the IRs were 99.61%, 99.19%, and 99.71%, respectively, compared with the mo-P-Group on day 4 (Figure 5f). Although sufficient phosphorus was added to the co-P-Group, the ANOVA results showed no significant difference in the algal cell densities between the three co-culture groups during the experiment.

The above results once again verified that the existence of *E. crassipes* inhibits the growth of *R. raciborskii*. In addition, the above results indicated that under experimental conditions, the supplementation of phosphorus, nitrogen, potassium, calcium, and trace elements could not promote the growth of *R. raciborskii* or alleviate the inhibitory effect of *E. crassipes* on *R. raciborskii*.

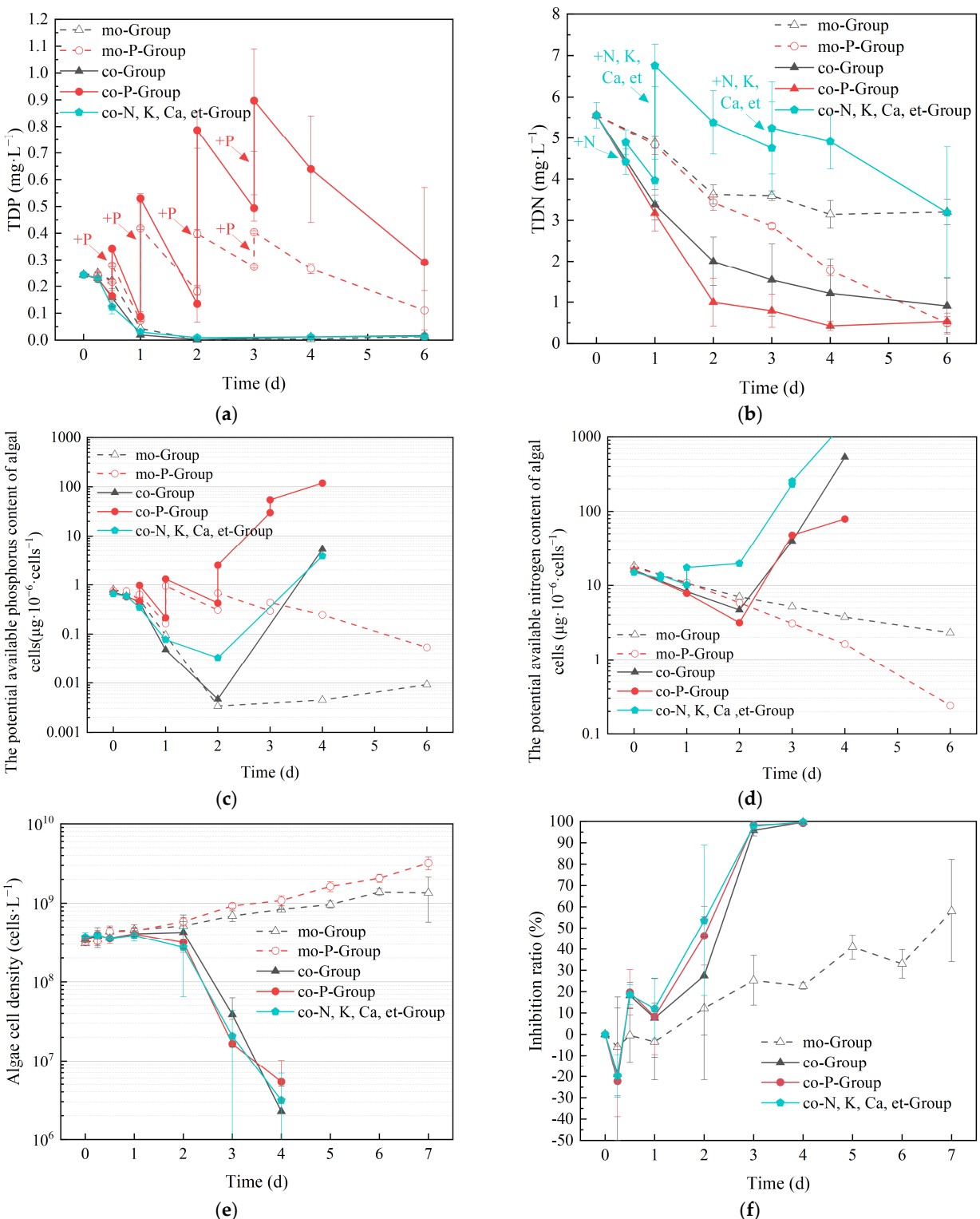

**Figure 5.** The concentrations of (**a**) TDP and (**b**) TDN in the culture medium, the potentially available amount of (**c**) phosphorus and (**d**) nitrogen obtained by a single *R. raciborskii* cell of each group, (**e**) the algal cell density, and (**f**) inhibition rate of *R. raciborskii* of each group while adjusting the addition of nutrients during culturing.

### 3.3. Effect of E. crassipes Exudates on the Growth of R. raciborskii

When incubated in *E. crassipes* exudates, the algal cell density of *R. raciborskii* was significantly lower than in the control group on day 5 and day 7 ($p < 0.01$; Figure 6). The IR

of *E. crassipes* exudates on *R. raciborskii* gradually increased, reaching 36.29% on day 7, and then decreased to 12.35 % on day 9 (Figure 6).

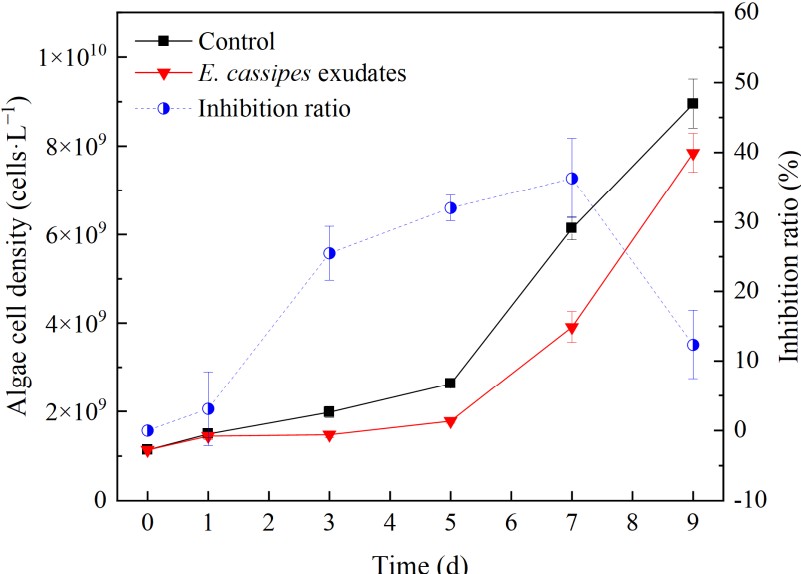

**Figure 6.** Algal cell density and inhibition ratio of *R. raciborskii* inoculated with *E. crassipes* exudates (mean ± SD, *n* = 3).

## 4. Discussion

The phenomenon in which *R. raciborskii* proliferates and becomes the dominant algae in the world has aroused widespread concern. However, the current research mainly focuses on exploring the competitive advantages of *R. raciborskii* and the reasons for its spread [4,7,11]. The research on control methods still needs to be improved. We chose *E. crassipes*, an aquatic macrophyte commonly used in aquatic ecological restoration, to explore the ecological control method of *R. raciborskii*. According to the results of co-culture experiments, we found that the cell density of *R. raciborskii*, with an initial cell density of $1.57 \times 10^8$ cells·L$^{-1}$ to $1.57 \times 10^9$ cells·L$^{-1}$, began to decrease after 3~5 days of co-culture with *E. crassipes*. The IR was up to 99% compared with the control group cultured alone. We also found that *E. crassipes* inhibited the lengthening of *R. raciborskii* trichomes to some extent. Previous studies have found that the trichome length of *R. raciborskii* is positively correlated with the concentration of nutrients in the environment [10,49,50]. Shorter trichomes have a larger specific surface area and can achieve more effective nutrient absorption, which may be why trichomes tend to become shorter when nutrients are insufficient [49]. Because of *E. crassipes*, most of the phosphorus in the culture medium was consumed and absorbed. Nutrient deficiency may inhibit the lengthening of *R. raciborskii* trichomes in co-culture groups. Based on these phenomena, we initially considered that nutrient competition may be the cause of inhibition.

Aquatic macrophytes and phytoplankton are primary producers in lake aquatic ecosystems, and naturally, they have a competitive relationship. *E. crassipes* has excellent nutrient absorption capacity, and nutrients are key factors in regulating algae growth. In the past, many scholars believed that nutrient competition was an essential mechanism for *E. crassipes* to inhibit algae growth [19,51,52]. When nutrient competition is a factor affecting the interspecific relationship, the regulation of nutrients affects the process of interspecific competition and even the outcome of interspecific competition [43,53]. However, the regulation of nutrients did not produce the expected results in the co-culture experiments. When *R. raciborskii* was cultured alone, the supplementation of phosphorus promoted an increase in algal cell density, indicating that phosphorus was one of the limiting factors in the growth of *R. raciborskii* under the initial nutrient conditions set in the experiment. The nutrient

factor can significantly affect the growth of *R. raciborskii* when cultured alone. However, when co-cultured with *E. crassipes*, phosphorus supplementation could not promote the growth of *R. raciborskii* or alleviate the inhibitory effect of *E. crassipes*, indicating that factors other than phosphorus restricted the growth of *R. raciborskii*. The result of supplying nitrogen, potassium, calcium, and trace elements was consistent with supplying phosphorus, indicating that nitrogen, potassium, calcium, and trace elements were not limiting factors. The effect of the nutrient factor was changed by the presence of *E. crassipes*. Following the above discussion, nutrient competition is actually not the primary mechanism by which *E. crassipes* inhibits the growth of *R. raciborskii*. Amorim et al. found that the presence of submerged macrophytes can significantly reduce cyanobacterial biomass, even under additional nutrient conditions, which is similar to the results of this study [54].

Floating-leaved macrophytes mainly inhibit algae growth through nutrient competition, shading, and allelopathy [51,55,56]. Clarifying the role of these mechanisms in practice remains a significant challenge. In this study, the shade factor was excluded by simulating the effect of *E. crassipes* shading on individually cultured *R. raciborskii*. This study did not investigate the inhibitory effect of shading in depth, mainly because *R. raciborskii* is considered highly shade-tolerant and can adapt to low-light conditions by adjusting its morphology [12,57]. Therefore, shading may not be an effective method for controlling *R. raciborskii* blooms. The present study also verified that *E. crassipes* has inhibitory allelopathy on *R. raciborskii*. Nevertheless, the inhibitory effect of *E. crassipes* exudates on *R. raciborskii* was not sustainable in the experiment. This phenomenon may be because the experiment failed to simulate the continuous release of allelochemicals from *E. crassipes*. Therefore, the allelochemical concentration was low and may have degraded gradually during cultivation [58]. Previous studies have shown that multiple dosing of allelochemicals has a significantly stronger inhibitory effect than one-time dosing [59]. After excluding shading and nutrient competition, allelopathy may be the primary mechanism by which *E. crassipes* inhibits the growth of *R. raciborskii*. Allelochemicals released by aquatic plants can damage microalgae cells through various pathways, including interfering with photosynthesis, generating oxidative stress, and damaging the cell membrane structure [19,60]. Trichome length is also affected by metabolic activity and cell division rate [61]. In this study, the inhibition of trichome elongation may be related to the allelopathy of *E. crassipes*, but the specific mechanism needs further study. Although nutrient competition is not the main mechanism of the inhibition of *E. crassipes*, the availability of nutrients affects the production of allelochemicals. However, different species show different results in terms of whether the ability of allelopathic inhibition is stronger when nutrients are abundant or limited [62–64].

While macrophytes affect the growth of algae, algae also affects the growth of macrophytes. For example, the fresh weight, plant height, and photosynthetic activity of *Potamogeton pectinatus* were inhibited when co-cultured with *M. aeruginosa*, with a density of more than $10^{10}$ cells·$L^{-1}$ [65]. Other studies have found that *M. aeruginosa* can inhibit the photosynthesis of *Vallisneria spiralis* by releasing allelochemicals [66]. In this experiment, the growth and photosynthesis of *E. crassipes* were not inhibited when co-cultured with *R. raciborskii* from $1.57 \times 10^7$ cells·$L^{-1}$ to $1.57 \times 10^9$ cells·$L^{-1}$, which demonstrated that *E. crassipes* is tolerant to *R. raciborskii* at this cell density. In addition, similar to the water purification effect of other phytoremediation plants, *E. crassipes* can improve water quality, such as stabilizing pH to neutral and reducing conductivity in this study [67].

Given the above, *E. crassipes* has a promising application for controlling *R. raciborskii* blooms. However, in view of the invasiveness of *E. crassipes*, we suggest that it should be strictly managed in practical engineering applications in areas where it is an alien species. The application of *E. crassipes* in lakes and reservoirs is relatively controllable compared to rivers. At present, *E. crassipes* is generally prevented from escaping by setting up fences and raising in pens, and it is prevented from being excessively propagated or decomposed to cause water quality deterioration by being periodically salvaged and harvested [28,31,68]. If *E. crassipes* spreads carelessly, it needs to be removed as soon as possible using mechanical

removal or chemical or biological reagents, according to the situation on the spot [69]. The controlled use of *E. crassipes* to remediate eutrophic lakes with *R. raciborskii* blooms can be investigated in the next step. In addition, this study also pointed out that allelopathy may be the primary mechanism by which *E. crassipes* inhibits the growth of *R. raciborskii*. The contribution and significance of other mechanisms, such as allelopathy, could be further discussed to control *R. raciborskii* blooms more reasonably and efficiently.

## 5. Conclusions

The present study demonstrated that under co-culture conditions, *E. crassipes* had an apparent inhibitory effect on *R. raciborskii*, with an initial cell density of $1.57 \times 10^8$ cells·L$^{-1}$ to $1.57 \times 10^9$ cells·L$^{-1}$, which could effectively inhibit the increase of algal cell density and trichome length. In contrast, the growth status and photosynthesis of *E. crassipes* plants were not inhibited. In the future, *E. crassipes* can be used to restore lakes with *R. raciborskii* blooms. In addition, in this study, the supplementation of phosphorus, nitrogen, potassium, calcium, trace elements, and other nutrients did not alleviate the inhibition of *E. crassipes* on *R. raciborskii*, indicating that a mechanism other than nutrient competition played a leading role. Allelopathy may be the primary mechanism by which *E. crassipes* inhibits the growth of *R. raciborskii*. In the next step, it is necessary to study the contribution and significance of allelopathy and other inhibitory mechanisms of *E. crassipes* to control *R. raciborskii* blooms more reasonably and efficiently by using *E. crassipes* and other aquatic macrophytes.

**Author Contributions:** Conceptualization, W.P. and X.C.; methodology, X.C.; software, X.C.; validation, X.C., Y.H. and Y.Z.; formal analysis, X.C. and X.H.; investigation, X.C.; resources, W.P.; data curation, X.C.; writing—original draft preparation, X.C.; writing—review and editing, X.C., Y.Z., Y.H. and W.P.; visualization, X.C.; supervision, W.P.; project administration, W.P.; funding acquisition, W.P. All authors have read and agreed to the published version of the manuscript.

**Funding:** This research received no external funding.

**Institutional Review Board Statement:** Not applicable.

**Informed Consent Statement:** Not applicable.

**Data Availability Statement:** The data in this study is available from the corresponding author on reasonable request.

**Conflicts of Interest:** The authors declare no conflict of interest.

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
