# Peer review of "Preliminary Study on the Inhibitory Effect and Mechanism of Eichhornia crassipes on Co-Cultured Raphidiopsis raciborskii"

_water, doi:10.3390/w15091690_

Round 1

Reviewer 1 Report

Review for the paper "Preliminary Study on the Inhibitory Effect and Mechanism of Eichhornia Crassipes on Co-cultured Raphidiopsis raciborskii" by Xiaowei Cheng, Weibin Pan, Yuanyuan Hu, Yulin Zou and Xiaojia Huang submitted to "Water".

General comment.

The present paper aimed to study the applicability of using the aquatic weed Eichhornia crassipes to control populations of harmful cyanobacteria Raphidiopsis raciborskii under experimental conditions. Harmful algal blooms play a significant role in aquatic ecosystems leading to increased mortality of the most biotic communities and thus reducing local fishery potential and water quality. Therefore, the research appears to have significance in the management of freshwater ecosystems and may be attractive to relevant scientists. The authors concluded that the addition of E. crassipes led to a decreased abundance of R. raciborskii suggesting its negative impact. They proposed that allelopathy might be the factor influencing a decline in R. raciborskii density. Methods and experimental design look appropriate. Statistical treatments are done correctly and allowed the authors to provide valid results. Illustration of the main findings is done in a clear manner. There are some minor issues to be addressed before the final acceptance.

Specific remarks.

L13-14. Consider replacing "may expand with climate warming" with "may expand its range with climate warming".

L23. Consider replacing "and the growth" with "but the growth".

L93. Provide coordinates for the pond.

L94-95. Consider replacing "Brushed the roots and leaves of the E. crassipes with a soft brush and repeatedly rinsed with tap water to remove the attached algae and sediment" with "The roots and leaves of the E. crassipes were brushed and repeatedly rinsed with tap water to remove the attached algae and sediment".

L97. Consider replacing "Replace the nutrient solution and prune withered roots and leaves every ten days" with "The nutrient solution was replaced and withered roots and leaves were cut off every ten days".

L130. Consider replacing "When R. raciborskii bloom" with "During R. raciborskii bloom".

L140, 174. Explain why the duration of the experiment was seven day.

L262. Consider replacing "co-culture" with "co-cultured".

Reviewer 2 Report

The manuscript presents the effect of cultivating water hyacinth in places where there is dangerous growth of certain cyanobacteria that are dangerous because of their toxin production. The work is very interesting and applied to the problem of the presence of this dangerous toxin-producing phytoplankton.

The introduction is correct and well presented. In my opinion it is necessary to present in the introduction (and not at the end of the discussion) also that E. crassipes is a plant that should be used with care because of its great power of growth, expansion and invasion of aquatic ecosystems, displacing other aquatic plants and invading the ecosystem in a way that can cause serious problems in certain places. For example, in many areas of Europe it is considered an invasive plant that should be exterminated in rivers and lakes. These aspects should be indicated, therefore, that the use of the hyacinth is only applicable in places where it is native or where it is exotic only under controlled conditions where it is not released into the environment.

The methodology is well described, but in experiments 1, 2 and 3 there are no academic references to the methodology used; if these are procedures and materials devised by the authors, they should indicate this; if this is not the case, they should indicate the documentary sources where applicable.

They should also explain why the time of the experiment is limited to seven days and why it has not been tested over a longer period.

As for the results, in my opinion there is a flaw in the experiment, and that is that P is taken up by the plants in their growth and they deplete the nutrient, which does not happen with nitrogen. This lack of P may condition the result obtained, since I consider that a P concentration similar to the initial one should have been maintained during the experiment, so that the result of the growth or not of the cyanobacteria would not have been due to the lack of P. This aspect should be pointed out in the results and presented in the discussion, as one of the weak points of the study. In a natural ecosystem, where eutrophication is constant, there will never be a P limitation since the contribution will be constant over time.

As for the bibliography presented, it is correct and is presented in accordance with the style of the journal.

Reviewer 3 Report

In the manuscript “Preliminary Study on the Inhibitory Effect and Mechanism of Eichhornia Crassipes on Co-cultured Raphidiopsis raciborskii” the authors observed the interaction between E. crassipes and R. raciborskii to verify whether E. crassipes can effectively inhibit the growth of R. raciborskii and further explore the inhibition mechanism of E. crassipes. This manuscript is well organized, and the drawn conclusions are coherent with the obtained results. The references should be updated to include more recent studies. 

Lines 34 – 35: The keywords should be arranged alphabetically.

Lines 375 – 457: The discussion should be expand adding other comments also on other study already published.

Round 2

Reviewer 2 Report

The authors have made important modifications to the content of the manuscript following the proposals presented in my review; some details of correctness of the English language are observed that do not seem adequate and could possibly be revised in the editing process.

As for the final content, in my opinion it is correct and can be published.